# Cerebrospinal Fluid System Infection in Children with Cancer: A Retrospective Analysis over 14 Years in a Major European Pediatric Cancer Center

**DOI:** 10.3390/antibiotics11081113

**Published:** 2022-08-17

**Authors:** Antonia Diederichs, Evelyn Pawlik, Anke Barnbrock, Stefan Schöning, Jürgen Konczalla, Tobias Finger, Thomas Lehrnbecher, Stephan Göttig, Konrad Bochennek

**Affiliations:** 1Division for Pediatric Hematology and Oncology, Hospital for Children and Adolescents, Johann Wolfgang Goethe-University, D-60596 Frankfurt, Germany; 2Institute for Medical Microbiology and Infection Control, Johann Wolfgang Goethe-University, D-60596 Frankfurt, Germany; 3Department of Neurosurgery, Johann Wolfgang Goethe-University, D-60596 Frankfurt, Germany

**Keywords:** hydrocephalus, cerebrospinal fluid system, infection, child, cancer

## Abstract

Infection of a cerebrospinal fluid system is a serious medical complication. We performed a retrospective monocentric analysis on temporary and permanent cerebrospinal fluid devices in children with and without cancer, covering a period of over 14 years. Between 2004 and 2017, 275 children with a cerebrospinal fluid system were seen at our institution. Thirty-eight children suffered from 51 microbiologically proven infectious episodes of the cerebrospinal fluid system (12 children with cancer and 26 children without cancer). Independently of the cerebrospinal fluid system used, the incidence of infection did not significantly differ between children with and without cancer and was the highest in children younger than one year. Infection occurred earlier in external ventricular drain (EVD) than ventriculoperitoneal (VP) shunt, and in EVD significantly earlier in children with cancer compared with patients without cancer. The pathogens isolated were mainly Gram-positive bacteria, in particular *Staphylococcus* spp., which should be taken into account for empirical antimicrobial therapy.

## 1. Introduction

Temporary or permanent cerebrospinal fluid systems using an external ventricular drain (EVD) or an internal system such as a ventriculoperitoneal (VP) shunt, ventriculoatrial (VA) shunt, or Rickham reservoir may be indicated in patients with hydrocephalus [1]. There are a number of causes for hydrocephalus, which depend on the age and underlying disease, and may include hemorrhage, infection, tumor, and congenital problems such as aqueduct stenosis or myelomeningocele [2,3]. Unfortunately, the implantation of these systems can cause complications such as infection, obstruction, mechanical shunt failure, and overdrainage, which depend on the system used and may be associated with significant morbidity and mortality [2,4,5,6,7].

Immunocompromised patients such as those undergoing therapy for cancer are at an increased risk for infectious complications. These infections may present as fever of unknown origin (FUO), and microbiologically or clinically documented infections (e.g., bacteremia, pulmonary infections, and soft tissue infections). The increased risk for infectious complications in patients with cancer can be explained by a number of alterations of the immune system, which are caused by the underlying malignancy and/or by treatment modalities such as chemotherapy, surgery, or irradiation. For example, in pediatric cancer patients, the skin and mucosa are often damaged, there is a loss of phagocytes, and cellular and humoral immunity are impaired [8]. It is well known that up to two-thirds of bacteremia in children with cancer are caused by Gram-positive pathogens [9,10], but there is a paucity of data of cerebrospinal fluid system infection in this patient population. We therefore analyzed the characteristics of pathogen proven cerebrospinal fluid system infections in pediatric cancer patients and compared the data to those in non-immunocompromised children, who do not exhibit additional risk factors for infectious complications such as neutropenia or mucositis.

## 2. Results

Overall, 275 children with an inserted cerebrospinal fluid system were identified who were seen in our hospital between 2004 and 2017 for different reasons (e.g., regular neurological controls or complications) (Figure 1). In 38 of these children, a total of 51 infectious episodes occurred, whereas 204 patients did not fulfill the criteria for proven cerebrospinal fluid system infection. Thirty-three patients were not included in the analysis as they were older than 18 years, were transferred to another institution during therapy for cerebrospinal fluid system infection, or their documentation of clinical data was insufficient. All but four permanent cerebrospinal fluid systems had been removed after diagnosis of infection. No patient died due to infection of the cerebrospinal fluid system.

### 2.1. Patients Characteristics and Cerebrospinal Fluid Systems

#### 2.1.1. Patients with Infection of the Cerebrospinal Fluid System

Thirty-eight children (25 males and 13 females) with a median age of one year and two months (range of 0 to 18 years) suffered from 51 microbiologically proven infectious episodes, which occurred in 50 different cerebrospinal fluid systems. In children aged under one year, infection of the cerebrospinal fluid system occurred in 14.4%, whereas this was seen in 9.9% of children older than one year. Twelve patients (nine males and three females) suffered from cancer and 26 children (16 male and 10 female) suffered from other underlying medical problems. Overall, the incidence of infection was 15.7%, and did not significantly differ between children with cancer [12.7% (12/94)] and children without cancer (17.6% (26/148)) (*p* = 0.3652).

Twenty-one children with an infected cerebrospinal fluid system had a VP shunt, twelve children an EVD, three a Rickham reservoir, and two a VA shunt.

The cerebrospinal fluid system in patients with infection differed between children with cancer and children without cancer. Eighteen children without cancer and an infected cerebrospinal fluid system had a VP shunt (69.2%), five an EVD (19.2%), two a VA shunt (7.7%), and one a Rickham reservoir (3.9%). In contrast, three children (25%) with cancer and an infected cerebrospinal fluid system had a VP shunt, seven an EVD (58.3%), and two a Rickham reservoir (16.7%) (Figure 2). When comparing the occurrence of infection in different cerebrospinal fluid shunt systems between children with and without cancer, no significant difference was found for the VP shunt systems (*p* = 0.4293) or EVD systems (*p* = 0.0508).

In total, in 16 of 38 patients with infection of the cerebrospinal fluid system, hemorrhage (among them, 11 patients with intraventricular hemorrhage) was the underlying medical problem, in twelve patients a tumor, in six a malformation and in four children other problems. Oncological diagnoses included ependymoma (*n* = 4), pilomyxoid astrocytoma (*n* = 3), medulloblastoma (*n* = 2), and acute lymphoblastic leukemia with involvement of the central nervous system, pontine glioma, and rhabdomyosarcoma with cerebral metastasis (each *n* = 1). Six out of the twelve children with cancer received intravenous chemotherapy, and two of them additionally received intrathecal administration.

In children with EVD, the most frequent underlying disease was a tumor (*n* = 7; 39%), followed by cerebral bleeding without intraventricular hemorrhage (*n* = 4; 22%) and congenital malformation (*n* = 4; 22%). In turn, children with permanent cerebrospinal fluid systems suffered from intraventricular hemorrhage (*n* = 12; 43%), followed by congenital malformation (*n* = 8; 29%) and tumor (*n* = 4; 14%) (Figure 3).

#### 2.1.2. Children without Infection of the Cerebrospinal Fluid System

Out of the 204 patients without microbiologically proven cerebrospinal fluid system infection, 82 patients had an underlying malignancy (26 females and 56 males), whereas 122 patients (53 females and 69 males) had other medical problems requiring a cerebrospinal fluid system. In this cohort, malignancies included astrocytoma (*n* = 33), medulloblastoma (*n* = 17), ependymoma (*n* = 6), low-grade glioma (*n* = 6), glioblastoma (*n* = 5), atypical teratoid rhabdoid tumor (*n* = 4), craniopharyngioma (*n* = 2), and various other malignancies (*n* = 9). Patients with other medical problems suffered from posthemorrhagic (*n* = 45) or postinfectious hydrocephalus (*n*= 11), congenital malformation (*n* = 58), and various other problems (*n* = 8). 

In children without infection of the cerebrospinal fluid system, the systems consisted of VP shunts (*n* = 140; 68.6%), external ventricular drainages (*n* = 43; 21.1%), and Rickham reservoirs (*n* = 21; 10.3%). In children with cancer and without infection of the cerebrospinal fluid system, EVD was inserted in 39 patients (47.6%), VP shunt in 32 (39%), and Rickham reservoir in 11(13.4%) (Figure 4). Eleven out of 83 pediatric cancer patients (13.3%), in whom no infection of the cerebrospinal fluid system was diagnosed, had received intrathecal chemotherapy.

### 2.2. Time Period between Implantation of the Cerebrospinal Fluid System and Infectious Complication

The time period between implantation of the cerebrospinal fluid system and the infectious episode ranged between zero and ten years (median of 18 days). The median time of the occurrence of infection after surgery was significantly shorter in EVD (*n* = 18) compared with VP shunt (*n* = 28; 11 days vs. 19.5 days; *p* = 0.0404). In addition, the median time between implantation and infection of EVD was significantly shorter in children with cancer compared with children without cancer (7 days; range, 6 to 19 days) vs. 14 days (range, 6 to 43 days; *p* = 0.0205).

Thirty-six (70.6%) infectious episodes occurred during the first 30 days after implantation of the cerebrospinal fluid system, 43 (84%) during the first 100 days, and 46 (90%) within the first year.

### 2.3. Characteristics of Infectious Episodes

The 51 infectious episodes occurred in 12 children with cancer (13 episodes) and in 26 children without a malignancy (38 episodes). At the time of the infectious episode, 22 patients were younger than 12 months, 7 between 13 and 24 months, and 22 older than 24 months. In 18 of the infectious episodes, patients had an EVD, in three episodes patients had a Rickham reservoir, in 28 episodes patients had a VP shunt, and in two episodes patients had a VA shunt.

Analyzing the children with cancer and an infectious episode of the cerebrospinal fluid system, seven patients (53.9%) had an EVD, four patients (30.7%) a VP shunt, and two a Rickham reservoir (15.4%) (Figure 5). There was a significant lower rate of infectious episodes of EVD in children with cancer (*n* = 7/46 infected, 15%) compared with those without a malignancy (*n* = 5/9 infected, 55%; *p* = 0.0002).

Ventriculitis was diagnosed in 49 episodes, and in two patients who suffered from peritonitis the pathogen was isolated among others in the shunt material (*Staphylococcus aureus* and *Staphylococcus capitis*, respectively). The duration of an infectious episode ranged from 6 to 57 days, with a median of 23 days. No difference was observed between children with and without cancer.

Clinical symptoms varied widely and were mostly uncharacteristic. Symptoms included fever (*n* = 28), followed by severely reduced general condition (*n* = 10); nausea and vomiting (*n* = 6); reduced vigilance (*n* = 5); abdominal discomfort (*n* = 4); seizures (*n* = 3); insufficient fluid uptake (*n* = 3); septic illness (*n* = 2); headache (*n* = 2); agitation (*n* = 2); increased oxygen demand (*n* = 2); and others such as anisocoria, bradycardia, arterial hypertension, or tight fontanel (each *n* = 1). No difference was observed between children with and without cancer.

### 2.4. Distribution of Pathogens

A total of 63 pathogens were isolated in 51 infectious episodes, which occurred in 50 different cerebrospinal fluid systems. In 42 of these episodes (82.3%) one pathogen was identified, in six episodes (11.8%) two pathogens, and in three episodes (5.9%) three pathogens were diagnosed.

Most of the pathogens were isolated from cerebrospinal fluid (*n* = 45) and/or from removed shunt material (*n* = 22). Pathogens were additionally isolated from intraoperative smears (*n* = 5), wound swabs (*n* = 6), blood culture (*n* = 1), or abscess cavity (*n* = 1). In 30 infectious episodes, the pathogen was detected in one specimen; in 21 episodes the pathogen was isolated in two or more specimens.

Fifty-five out of 63 infections (87.3%) were caused by Gram-positive pathogens, mostly by *Staphylococcus* spp. (*n* = 42/55, 76.36%), which included *coagulase-negative staphylococci* (*n* = 35), of which *S. epidermidis* was the most prevalent species (*n* = 20, 36.36%), and *S. aureus* (*n* = 7, 12.7%) (Table 1).

Gram-negative pathogens were isolated in eight episodes (12.7%), of which six of them were in the EVD.

The distribution of Gram-positive and Gram-negative pathogens significantly differed between the various cerebrospinal fluid systems. Among the 33 pathogens isolated in VP shunt devices, one Gram-negative pathogen was found, whereas in children with an EVD, Gram-negative pathogens were isolated significantly more often (1/33 versus 6/24, *p* = 0.0161). In contrast, *S. epidermidis* was significantly more often isolated in a VP shunt system than in an EVD (13/16 vs. 3/16, *p* = 0.0369). 

In children with cancer, 17 pathogens were isolated in 12 patients suffering from 13 episodes of infection. In four patients, more than one pathogen was detected in one episode. In these patients, all isolated pathogens were Gram-positive (*n* = 17) and included 13 (76.5%) *Staphylococcus* spp., mostly *S. epidermidis* (*n* = 6; 35.3%). In children without cancer, 46 pathogens were isolated in 26 patients during 38 episodes of infection. In these children, 82.6% of the infections were caused by Gram-positive pathogens (*n* = 38) and 17.4% by Gram-negative pathogens (*n* = 8). The distribution of Gram-positive pathogens did not statistically differ between children with and without cancer (*p*= 0.6386).

Among the Gram-positive pathogens, all staphylococci were susceptible to vancomycin and linezolid. Resistance towards oxacillin, fosfomycin, and rifampicin was species specific; while all *S. aureus* were tested susceptible, resistance rates for CoNS were >40% for oxacillin and fosfomycin (Figure 6). *Staphylococcus* spp. of cancer patients were consistently more susceptible and exhibited less resistance towards the antibiotics tested.

### 2.5. Treatment

After a diagnosis of infection, 46 of 50 cerebrospinal fluid systems had been removed.

In all infectious episodes, intravenous antibiotic therapy was administered, which was supplemented with intrathecal antibiotics in 14 episodes. No adverse event of grade 3/4 was observed due to the administration of antibiotics.

For systemic treatment, 28 different antimicrobial medications were used, specifically, 24 antibiotic, 3 antifungal, and 1 virostatic compounds. The median number of antimicrobial compounds was three, with a maximum of up to 12 antimicrobial compounds per patient administered during therapy. Therapy was started in 30 patients as combination therapy, and in 21 patients as monotherapy, mainly with a cephalosporin antibiotic.

The duration of treatment did not significantly differ between patients with an EVD (*n* = 18, 23.5 days) and those with a permanent cerebrospinal fluid system (*n* = 33, 23 days; *p* = 0.9764). Similarly, no difference in treatment duration was seen between children with cancer (median 23 days) and children without cancer (median 23 days; *p* = 0.544). In seven of the children with cancer, infection of the cerebrospinal shunt system delayed chemotherapy and/or radiation (median 19 days, range 9–30 days).

## 3. Discussion

Cerebrospinal fluid systems are needed for many patients suffering from hydrocephalus, and infection is a potential complication of these devices [11,12,13]. In our analysis, the overall incidence of infections in cerebrospinal fluid systems was 15.7%, which is in line with the reported incidences in other studies [5,14,15,16,17,18].

Notably, according to the literature, the incidence of infectious complications of cerebrospinal fluid system is considerably higher in children compared with adults [7,16,19,20]. In this respect, children under one year of age seem to be at a particular high risk for infectious complications [2,3,4,6,7,21,22,23,24,25,26,27], which corroborates our findings. 

In our study, the incidence of infections of the cerebrospinal fluid system in children with cancer was not significantly different to that in children without cancer. Various rates of infection of the cerebrospinal fluid system in children with tumors have been reported in the literature, ranging from 0% to 23% [5,7,24,28,29,30]. This wide variation might be, at least in part, because studies used different definitions of an infectious episode. For our analysis, inclusion criteria required the isolation of a causative pathogen in addition to clinical symptoms and/or abnormalities. In contrast, other groups defined infection of the cerebrospinal fluid system by clinical symptoms and surrogate markers such as leukocytosis or elevated levels of the C-reactive protein, which are unspecific and may be pathological in other settings outside infection [12,22]. The fact that different cerebrospinal shunt systems are differently prioritized in individual institutions might also have accounted for the differences in the risk for infection. In this respect, in our analysis, most of the children with cancer had an EVD, whereas in other studies, the majority of patients had a VP shunt, Ommaya reservoir, or Rickham reservoir [7,24,28,29].

Overall, in our study, most children had a VP shunt implanted, and less often an EVD, a Rickham reservoir, or a VA shunt, respectively. Considerable differences were seen between children with and without cancer, with a higher percentage of VP shunts in children without cancer; however, the risk for VP shunt infection did not significantly differ between both groups. In contrast, EVD was seen considerably more often in children with cancer than in children without cancer. As little data of other groups have been published to date, a comparison of our data with the literature is difficult. 

Not surprisingly, we found that infections occurred significantly earlier in patients with EVD compared with those with a VP shunt. Although our data are comparable with the results of other authors [5,22,27,31], comparisons have to be interpreted with caution because of the small numbers of patients. 

Our data also indicate that children with cancer have a significantly lower risk of EVD infection compared with children without cancer. Importantly, the time from insertion until removal of the device, which is known to be one of the most important risk factors for infection [22], did not significantly differ between both patient groups (15 vs. 13 days). Therefore, one can only speculate whether the lower risk of infection might be due to the fact that physicians and/or nurses are especially cautious when treating children with cancer. Similarly, it is difficult to explain the fact that an infection of the EVD occurred significantly earlier in children with cancer than in children without cancer, which has not been reported before. Further studies are warranted to confirm our findings. 

Risk of infection and the distribution of pathogens varied and depended on the different cerebrospinal fluid systems. For example, 13% of patients with VP shunts had an infection, and staphylococci were isolated in two-thirds of the patients (Table 1). This indicates that skin flora causing secondary endogenous infections via the devices are the main route of infection. Data from antimicrobial resistance testing indicate that vancomycin, linezolid, and rifampicin are suitable for empirical treatment, whereas resistance rates for beta-lactams and fosfomycin were >40%. Surprisingly, pathogens of children with cancer showed reduced resistance to antibiotics compared with those in children without cancer. This is in contrast to previous reports [10,32], and may be explained by the fact that in children with cancer, infection of the shunt device occurred early after the diagnosis of the malignancy, and before the patient had received antibiotic agents for a longer period, increasing the risk for resistant pathogens. 

In 21.8% of patients with EVD infection occurred and in a significant proportion of these patients Gram-negative pathogens were isolated. Surprisingly, we did not observe Gram-negative pathogens in the cerebrospinal shunt system of children with cancer, although Gram-negative infections are common in children with cancer [33,34]. Our results corroborate data reported in the literature [14,25,26,30,33,34], where Gram-negative pathogens are more common in cerebrospinal fluid systems in children in the intensive care unit or in newborns [35].

Any infectious complication of a cerebrospinal fluid system may be associated with considerable morbidity, but in children with cancer, the infection may also lead to a delay in therapy, which in turn might negatively impact on the outcome. In our analysis, the infectious complication delayed chemotherapy for up to four weeks in seven out of twelve children with cancer, but the small number of patients does not allow for a solid statement regarding the impact of outcome. 

We recognize that our study has important limitations. For example, the fact that we performed a retrospective monocentric analysis prevents our data from being generalized to other institutions. On the other hand, we included in our analysis cerebrospinal fluid system infections in children with and without cancer, which allowed for a direct comparison of these two patient populations. In addition, we applied a stringent definition of an infection of a cerebrospinal fluid system, which prevents major bias.

In summary, our data show that cerebrospinal fluid system infections remain an important cause for morbidity in the pediatric setting, both in children with and without cancer. Our analysis also demonstrates that children with cancer do not have a higher risk of cerebrospinal fluid systems infection compared with children without cancer, and that most common pathogens are Gram-positive isolates, which should be taken into account for antimicrobial therapy. For optimizing antimicrobial therapy, these data, including data on pathogen resistance, need to be continuously monitored. 

## 4. Patients and Methods

We retrospectively reviewed medical charts including reports on imaging and surgery of all patients ≤18 years of age with an inserted cerebrospinal fluid system who were seen for different reasons (e.g., planned neurological controls or with complications) at our institution between January 2004 and December 2017. Clinical and microbiological data were retrieved using analysis databases Swisslab (Nexus), HyBASE (epiNET AG), and ORBIS (Dedalus). Extracted data included site of infection (e.g., ventriculitis, wound infection, or peritonitis), antibiotic therapy (compound, dosage, and duration of treatment), microbiological results (bacterial species identification and antimicrobial susceptibility testing), and surgical interventions such as explantation of the cerebrospinal fluid system. In addition, concomitant chemotherapeutics, given either intravenously and/or intrathecally, were documented. Antimicrobial resistance was determined employing clinical breakpoints version 8.0 and guidelines set by EUCAST (http://www.eucast.org/clinical_breakpoints/ accessed on 1 July 2018).

Cerebrospinal fluid system infection was defined by clinical symptoms and/or abnormalities (e.g., pleocytosis in the cerebrospinal fluid), in addition to the isolation of a causative pathogen in the cerebrospinal fluid or shunt material, respectively. The start and end of an infectious episode was defined as the initiation and cessation of antimicrobial treatment, respectively. Patients who were transferred to another institution during antimicrobial treatment or did not have sufficient data for the analysis were excluded from the analysis.

GraphPad Software, Inc. (San Diego, CA, USA), Prism for Windows, Version 5.04, was used for statistical analysis. Groups were compared using the Mann–Whitney U test. Statistical significance was defined as a *p*-value (two-tailed) < 0.05.

The study was approved by the local Ethics committee (440/18).

## Figures and Tables

**Figure 1 antibiotics-11-01113-f001:**
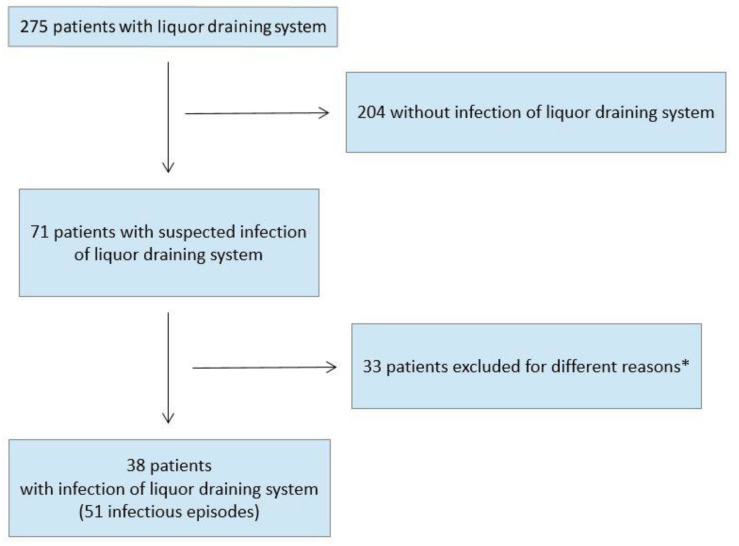
Flowchart of patient selection. * age older 18 years, transfered to another institution during therapy, or insufficient clinical data.

**Figure 2 antibiotics-11-01113-f002:**
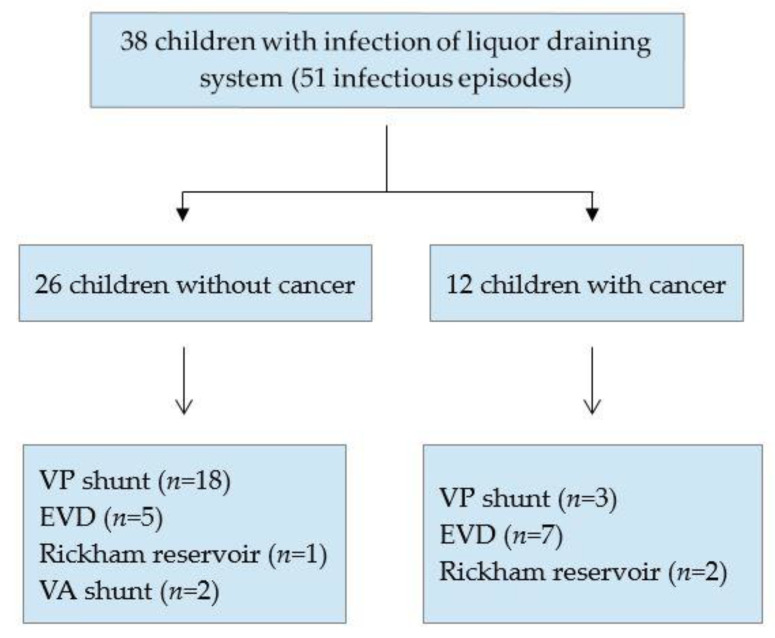
Overview of children with infection of a liquor draining system. EVD, external ventricular drain; VP, ventriculoperitoneal; VA, ventriculoatrial.

**Figure 3 antibiotics-11-01113-f003:**
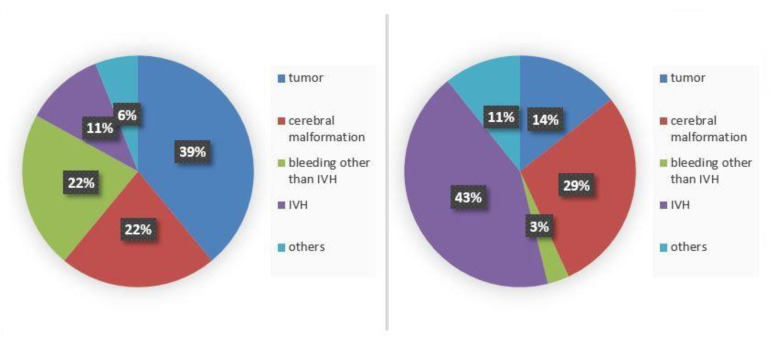
Underlying diseases in infected cerebrospinal fluid systems (EDV on the left (*n* = 22), other cerebrospinal fluid systems on the right (*n* = 39)). EVD external ventricular drain; IVH intraventricular hemorrhage.

**Figure 4 antibiotics-11-01113-f004:**
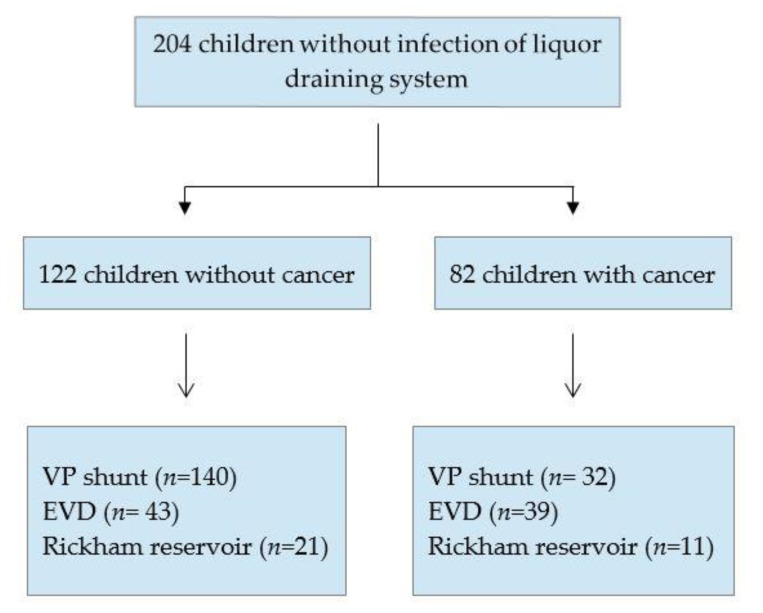
Overview of children without infection. EVD, external ventricular drain; VP, ventriculo-peritoneal.

**Figure 5 antibiotics-11-01113-f005:**
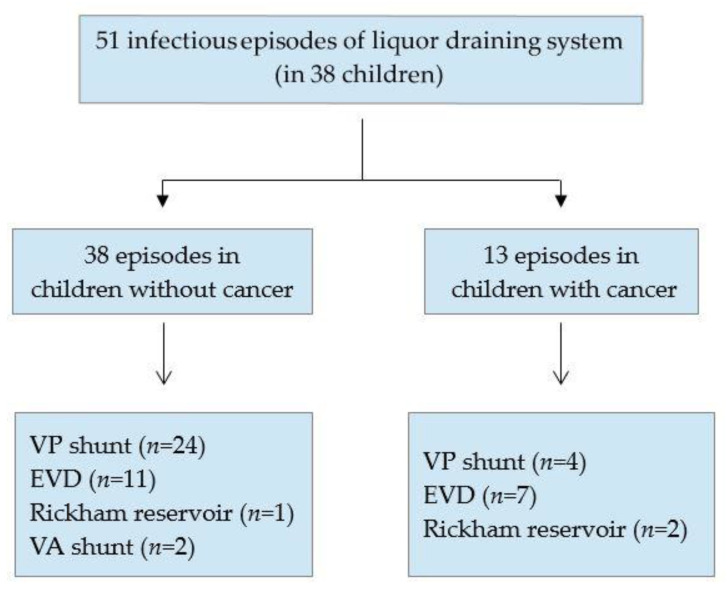
Overview of episodes of cerebrospinal fluid system infection. EVD, external ventricular drain; VP, ventriculoperitoneal; VA, ventriculoatrial.

**Figure 6 antibiotics-11-01113-f006:**
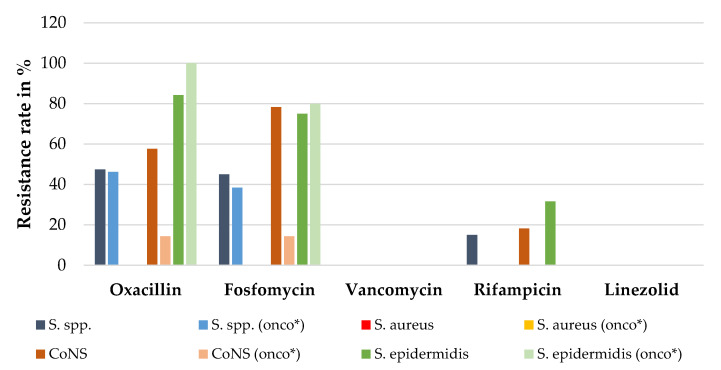
Resistance rates (in vitro) of *Staphylococcus* spp. isolated in children with cerebrospinal shunt systems against the most commonly used antibiotics in percentage, in general and for children with cancer (“onco*”) specifically.

**Table 1 antibiotics-11-01113-t001:** Distribution of pathogens isolated.

Pathogens Isolated (*n* = 63)	N	%
CoNS (*S. epidermidis*)	35 (20)	55.5 (31.7)
*Staphylococcus aureus*	7	11.0
*Bacillus* spp.	3	4.8
*Micrococcus luteus*	3	4.8
*Enterococcus faecalis*	2	3.2
*Corynebacterium* spp.	2	3.2
*Streptococcus* spp.	2	3.2
*Moraxella* spp.	2	3.2
*Enterobacter* spp.	2	3.2
Others ^1^	5	7.9

CoNS, coagulase-negative staphylococci. ^1^ Others: Paenibacillus spp., Acinetobacter baumannii, Klebsiella oxytoca, Pseudomonas aeruginosa, and Haemophilus influenzae (in each case n = 1).

## Data Availability

From the corresponding author upon request.

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
