# Peer review of "Cerebrospinal Fluid System Infection in Children with Cancer: A Retrospective Analysis over 14 Years in a Major European Pediatric Cancer Center"

_antibiotics, 2022, doi:10.3390/antibiotics11081113_

Round 1

Reviewer 1 Report

Single-center retrospective study as indicated by the authors presenting the casuistry of meningitis in pediatric patients with and without cancer. The authors should answer the following questions:

What was the mortality of the series?

Have there been complications or side effects with any antibiotic regimen?

According to the authors, children with cancer have fewer EVD infections, what do they think is the reason?

The paper should be reorganized: in the material and methods, point 4 has been erroneously included (it should be 2).

Author Response

Single-center retrospective study as indicated by the authors presenting the casuistry of meningitis in pediatric patients with and without cancer. The authors should answer the following questions:

What was the mortality of the series?

 The reviewer raised an important point – according to his/her suggestion, we included a statement that no patient has died due to the infection of the cerebrospinal fluid system (line 63 of the revised manuscript).

Have there been complications or side effects with any antibiotic regimen?

 According to the reviewer´s suggestion, we have included the statement „No adverse event of grade 3/4 was observed due to the administration of antibiotics“ (lines 217/218).

According to the authors, children with cancer have fewer EVD infections, what do they think is the reason?

 This is another excellent question by the reviewer, which we have extensively discussed in our group. We can only speculate whether the lower risk of infection might be due to the fact that physicians and/or nurses are especially cautious when treating children with cancer. We have included this statement in the revised version of the manuscript (lines 269-271).  

The paper should be reorganized: in the material and methods, point 4 has been erroneously included (it should be 2).

According to the structure given by the journal, the section “Patients and Methods” is located at the end of the manuscript, after the section results. We therefore have kept this organization.

Reviewer 2 Report

The current paper discussed the correlation of cerebrospinal fluid system infection in children with cancer in a retrospective analysis of 14 years and dissected the major pathogens that cause infection. There are some minor issues:

1. Introduction part could be expanded.

2. There are some typos, for example, line 87.

3. The authors could add tables or graphs to 2.1 through 2.3 to better present data. So far, it is a little bit confusing.

4. Are there any correlations between the cerebrospinal fluid systems the patients used and the distribution of pathogens?

Author Response

The current paper discussed the correlation of cerebrospinal fluid system infection in children with cancer in a retrospective analysis of 14 years and dissected the major pathogens that cause infection. There are some minor issues:

  1. Introduction part could be expanded.

According to the reviewer´s suggestion, we expanded the introduction part. Specifically, we focused on infectious complications in immunocompromised children, and included risk factors and various types of infectious complications (lines 33/34, 37, 40-47, 52-53). We also included additional references (ref #7-9).   

  1. There are some typos, for example, line 87.

We have corrected the typos including the typo mentioned by the reviewer.

  1. The authors could add tables or graphs to 2.1 through 2.3 to better present data. So far, it is a little bit confusing.

We absolutely agree with this fantastic idea by the reviewer and included Figures (Figures 2, 3, 4, and 5) in order better present our data.

  1. Are there any correlations between the cerebrospinal fluid systems the patients used and the distribution of pathogens?

The question raised by the reviewer is important. Although we have addressed these findings already in the original manuscript, we have rephrased this part for clarification (lines 190-195)